# Life Cycle Assessment and Impact Correlation Analysis of Fly Ash Geopolymer Concrete

**DOI:** 10.3390/ma14237375

**Published:** 2021-12-01

**Authors:** Xiaoshuang Shi, Cong Zhang, Yongchen Liang, Jinqian Luo, Xiaoqi Wang, Ying Feng, Yanlin Li, Qingyuan Wang, Abd El-Fatah Abomohra

**Affiliations:** 1Key Laboratory of Deep Earth Science and Engineering (Ministry of Education), Department of Architecture and Environment, Sichuan University, Chengdu 610065, China; wangqy@scu.edu.cn; 2Failure Mechanics and Engineering Disaster Prevention and Mitigation Key Lab of Sichuan Province, Sichuan University, Chengdu 610065, China; zhangcong@stu.scu.edu.cn (C.Z.); liangyc@stu.scu.edu.cn (Y.L.); luojinqian@stu.scu.edu.cn (J.L.); wangxqi0126@163.com (X.W.); phying@stu.scu.edu.cn (Y.F.); tliyanlin@163.com (Y.L.); 3Department of Mechanical Engineering, Chengdu University, Chengdu 610106, China; 4Department of Environmental Engineering, School of Architecture and Civil Engineering, Chengdu University, Chengdu 610106, China; abomohra@cdu.edu.cn

**Keywords:** geopolymer concrete, compressive strength, grey relational analysis, multivariate analysis of variance, CO_2_ emission

## Abstract

Geopolymer concrete (GPC) has drawn widespread attention as a universally accepted ideal green material to improve environmental conditions in recent years. The present study systematically quantifies and compares the environmental impact of fly ash GPC and ordinary Portland cement (OPC) concrete under different strength grades by conducting life cycle assessment (LCA). The alkali activator solution to fly ash ratio (S/F), sodium hydroxide concentration (C_NaOH_), and sodium silicate to sodium hydroxide ratio (SS/SH) were further used as three key parameters to consider their sensitivity to strength and CO_2_ emissions. The correlation and influence rules were analyzed by Multivariate Analysis of Variance (MANOVA) and Gray Relational Analysis (GRA). The results indicated that the CO_2_ emission of GPC can be reduced by 62.73%, and the correlation between CO_2_ emission and compressive strength is not significant for GPC. The degree of influence of the three factors on the compressive strength is C_NaOH_ (66.5%) > SS/SH (20.7%) > S/F (9%) and on CO_2_ emissions is S/F (87.2%) > SS/SH (10.3%) > C_NaOH_ (2.4%). Fly ash GPC effectively controls the environmental deterioration without compromising its compressive strength; in fact, it even in favor.

## 1. Introduction

In order to effectively reduce the negative impact of the construction industry on global warming, countries around the world are also actively responding to reduce energy consumption and emissions. In this context, China has put forward the goal of “reaching the carbon peak by 2030 and becoming carbon neutral by 2060”, demonstrating China’s responsibility to address global climate change actively. As an essential component of concrete materials, silicate cement is one of the most used building materials in modern construction. The industrial production of cement consumes many resources and energy, with the energy consumed reaching 10% of the total global energy consumption [1]. The calcination stage of raw materials emits a large amount of CO_2_ and other harmful gases, causing severe environmental pollution [2].

Geopolymer concrete (GPC) has been recognized as an ideal new environmentally friendly building material for the construction industry, reducing the use of energy-intensive, emission-intensive cement, and thus reducing the environmental impact to a certain extent. However, the extent to which geopolymers can reduce environmental impacts and the significance of their impact on various environmental impact indicators is unknown. Nowadays, life cycle assessment (LCA) is considered one of the most systematic and scientific-based environmental assessment tools for carrying out the environmental evaluation of building materials through the whole life cycle [3,4], as described in detail by IS014040. However, most LCA studies focus on the environmental impact of ordinary Portland cement (OPC) concrete or blended cement concrete [3,4,5,6]. It was found that the heavy environmental load of OPC concrete is mainly due to the high energy consumption and greenhouse gas emissions of the cement [7].

Few articles are reporting the LCA of GPC. For example, Turner [8] estimated that CO_2_ emissions of 1 m^3^ GPC from the mining source to produce concrete were 9% lower than OPC concrete. The metakaolin-based geopolymer could significantly decrease the CO_2_ emissions by 27–45% compared to the OPC concrete [9]. Alkali-activated binary concrete had an equal or even higher compressive strength than the OPC concrete and a clear environmental advantage as its carbon footprint was 44.7% lower [3]. The slight difference between them is the preparation of the alkali activators and the need for elevated temperature curing of geopolymer concrete to achieve reasonable strength. McLellan [10] conducted a detailed environmental assessment of the production of GPC in Australia. It can reduce greenhouse gas emissions by 44–64% compared to OPC. They also suggest that the benefits have more potential if the feedstock is sourced appropriately and low transport costs. Chen [11] also came to a similar conclusion. The “cradle to gate” model commonly used in LCA does not capture environmental impact beyond the gate and is only applicable when comparing GPC production with OPC [12]. Faridmehr [13] investigated the LCA of ternary blended AAM. According to the performance criteria, the boundary of the cradle to gate system is extended to include the mechanical and sulfate resistance of AAM. The modified LCA with respect to CS revealed the lower intensity of normalized CO_2_ emissions in the AAM mixture containing high-volume FA and GBF. For AAM mixtures containing POFA, its relatively low CS and high amount of electricity required for oven drying of POFA lead to the highest intensity of normalized CO_2_ emissions.

All these studies came to a similar conclusion that GPC has lower CO_2_ emissions than OPC concrete under a specific compressive strength. Almost all the quantitative studies of GPC were based on a single compressive strength without a comparative analysis of GPC and OPC concrete under different compressive strengths. Compressive strength was used as the only or primary standard for mix proportion design, ignoring the importance of the environmental impact. However, the combined effect of parameters on mechanical strength and the environment should be considered in green concrete design. Accordingly, this study has quantified and compared the environmental impact of fly ash GPC and OPC concrete under different strength grades from manufacturing to production and found which steps and materials have significant environmental impacts. Multivariate Analysis of Variance (MANOVA) and Gray Relational Analysis (GRA) were used to determine the main impact factors and obtain the correlation of different impact factors contributing to CO_2_ emissions and compressive strength. This study further demonstrates the advantages of GPC in carbon reduction and helps promote its application. Furthermore, it provides designers with a basis for designing mix ratios to produce concrete with adequate compressive strength and low environmental impact.

## 2. Materials and Methods

### 2.1. Materials and Sample Preparation

The aggregates used in the current study included coarse aggregates and fine aggregates. The fly ash was class F low calcium fly ash with an average diameter of 1.586 µm. The characteristics of aggregates (Appendix A) and fly ash (Appendix A) are provided in the Appendix A.

The alkali activator solution was composed of sodium silicate (Ms = 3.13, 27.64% SiO_2_, 8.83% Na_2_O) and sodium hydroxide solution, which was prepared one day before the experiment. Sodium hydroxide solution was prepared with 98% purity of sodium hydroxide in the laboratory. The concentration was 12 mol/L.

After all the materials were prepared, GPC and OPC of different strengths were designed based on our previous studies, and 100 × 100 × 100 mm specimens were prepared for compressive strength testing (Appendix A). Firstly, dry mix the fine and coarse aggregate in the mixer for 1 min; then, pour into the fly ash and mix for another 1 min. Secondly, add the alkali activator solution and continue mixing for 2 min. Finally, the fresh concrete is placed in steel molds and compacted on a vibrator. In order to promote the development of compressive strength, GPC was covered with plastic film to prevent moisture loss and cured at 80 °C for 24 h and then cured at room temperature for 27 days. OPC was cured for 28 days under standard curing conditions.

### 2.2. LCA Model of GPC

According to IS014040, the LCA method was divided into four steps: functional unit and system scope definition, life cycle inventory analysis (LCI), life cycle impact assessment (LCIA), and life cycle interpretation [14].

#### 2.2.1. Functional Unit and System Scope

This study’s functional unit and system scope can be described as cradle to gate, with the main processes of 1 m^3^ GPC from manufacturing and transportation to production, as shown in Figure 1. The system mainly includes three stages of raw material production, transportation, and concrete production. The sand factory produces the aggregates, the fly ash is from the power station, and the chemical plant produces an alkali activator. The preparation of concrete materials includes five steps from mixing to demolding.

The following equation was considered to estimate the CO_2_ emission per cubic meter of the GPC and OPC:(1)$${\mathrm{CO}}_{2}\;\mathrm{emission}=\sum_{i=1}^{n}m_{i}(p_{i})$$
where the left-hand side of the equation indicates the amount of CO_2_ emission (kg CO_2_/m^3^) for every cubic meter of concrete, *m_i_* indicates the fraction of component *i*, and *p_i_* specifies the CO_2_ emissions (kg).

#### 2.2.2. Life Cycle Inventory Analysis (LCI)

The LCI of GPC production is mainly based on the mix proportions (input of various raw materials and energy) and emissions. The mixed proportions of the fly ash GPC and OPC concrete under different strength grades are shown in Appendix A. The emission factors (Table 1 and Table 2) were adopted from the Chinese Life Cycle Database (CLCD) built in the eBalance software developed by the Yi Ke Environmental Technology Company and the Institute of Sustainable Consumption and Production Sichuan University. The studied environmental impacts mainly included:CO_2_ emissions of the raw materials production process.CO_2_ emissions of the transportation of raw materials.CO_2_ emissions of production: mixing, vibrating, and curing at 80 °C for 24 h.

Kawai [15] provided the CO_2_ emissions of mixing, vibrating, and curing under room temperature. Turner [8] calculated the CO_2_ emissions of curing under elevated temperature (approximately 16 h) at an average temperature of 50 °C, which was extrapolated to 24 h (plus about 9 h of gradual heating) as 39.97 kg CO_2_/m^3^.

Considering the thermal power stations and cement plants were generally located in the surrounding fixed area and few in number, fly ash and OPC’s transportation distance was assumed to be 300 km.

From the point of the European Directive, fly ash cannot be regarded as waste anymore and is a by-product. Chen [11] has put forward the economic value allocation procedure:(2)$$C_{e}=\frac{{\left(e\times m\right)}_{by-product}}{{\left(e\times m\right)}_{by-product}+{\left(e\times m\right)}_{major-product}}$$
where e×m represents the unit price of product multiplied by the mass of the products.

Li [16] reported that 0.399 kg hard coal is consumed and 0.109 kg fly ash is produced per kilowatt hour in a thermal power enterprise. The price per kilowatt hour was estimated as 0.53 Yuan in China, and the class F fly ash cost represents 150 Yuan per ton, including freight charges. Therefore, the environmental impact allocation coefficient of fly ash can be calculated and shown in Table 3. Thus, the carbon emission factor of fly ash is 2.280 × 10^−2^ kg CO_2_/m^3^.

#### 2.2.3. Life Cycle Impact Assessment (LCIA)

In the production process, the corresponding CO_2_ emissions were calculated according to the standard for measuring, accounting, and reporting carbon emissions from buildings (CECS 374:2014).

### 2.3. Gray Relational Analysis (GRA)

Gray Theory is a systematic scientific theory first pioneered by Professor Deng Julong. In the Gray Theory, gray relational analysis (GRA) is commonly used to evaluate the effect of parameters on the evaluation subject. It is based on the sample data of each factor and uses the gray relational degree to describe the correlation degree of all the different factors. If the changes of the two factors are basically consistent, the correlation degree between them is larger, which means the influence is more significant. The leading advantage of the gray model is that it is also applicable to come with few data or irregular data points for establishing a model.

The basic idea of GRA is that the original data of the evaluation index is firstly processed by the dimensionless method. Then, the correlation coefficient and correlation degree are calculated. Finally, the evaluation index is found according to the correlation degree. Based on the method and theory of GRA, a gray relational model was established with the below computing methods and procedures:The sequence matrix for the gray relational model:Reference sequence: $X_{0}=\left(x_{0}\left(1\right),\;x_{0}\left(2\right),\;\ldots ,\;x_{0}\left(n\right)\;\right);\;X_{1}=\left(x_{1}\left(1\right),\;x_{1}\left(2\right),\;\ldots ,\;x_{1}\left(n\right)\;\right);$Compare sequence: $X_{2},\;\ldots ,X_{i}$;
(3)$$X_{i}=\left(x_{i}\left(1\right),\;x_{i}\left(2\right),\;\ldots ,\;x_{i}\left(n\right)\right).$$Dimensionless processing of raw data:
(4)$$X_{i}^{’}=\frac{X_{i}}{x_{i}\left(1\right)}=\left(x_{i}^{’}\left(1\right),x_{i}^{'}\left(2\right),\;\ldots ,x_{i}^{'}\left(n\right)\right)$$Difference sequence:
(5)$${∆}_{i}=\left({∆}_{i}\left(1\right);\;{∆}_{i}\left(2\right);\ldots ;\;\ldots_{i}\left(n\right)\right);\;{∆}_{i}\left(k\right)=\left|x_{0}^{’}\left(k\right)-x_{i}^{’}\left(k\right)\right|$$Maximum and minimum of difference sequence:
(6)$$M=\begin{array}{c} max\\ i \end{array}\begin{array}{c} max\\ i \end{array}{∆}_{i}\left(k\right);m=\begin{array}{c} min\\ i \end{array}\begin{array}{c} min\\ i \end{array}{∆}_{i}\left(k\right).$$Gray relation coefficient:
(7)$$\delta_{0i}\left(k\right)=\frac{m+\rho M}{{∆}_{i}\left(k\right)+\rho M};k=1,2,\;\ldots ,\;n;i=1,2,\;\ldots ,\;m.$$
ρ: distinguishing coefficient; ρ∈ (0, 1), which is around 0.5.Gray relation degree:
(8)$$r_{i}\left(k\right)=\frac{1}{n}\sum_{k=1}^{n}\delta_{0i}\left(k\right);i=1,2,\;\ldots ,\;n.$$

## 3. Results

### 3.1. Interpretation and Comparison of GPC and OPC Concrete

Figure 2 compares and demonstrates the distribution of CO_2_ emissions in GPC and OPC. Figure 3 displays the distributions of the ingredients and different production processes of GPC and OPC, respectively.

Compared with the CO_2_ emissions of GPC and OPC concrete, the minimum emission of GPC with 40 MPa is 260.14 kg CO_2_/m^3^, and the maximum is 336.54 kg CO_2_/m^3^. However, OPC concrete with 40 MPa showed a variation from 334.90 to 412.58 kg CO_2_ /m^3^, which is higher than GPC. The highest CO_2_ emission was recorded in C60 and C70 of GPC (333.28 kg CO_2_/m^3^ and 305.38 kg CO_2_/m^3^, respectively), which is significantly lower than the minimum in OPC concrete with the same compressive strength (405.26 kg CO_2_/m^3^). CO_2_ emissions from GPC at 40 MPa, 60 MPa, and 70 MPa are reduced by 20.48%, 27%, and 34.6% respectively compared to OPC. It seems that the higher the compressive strength of the geopolymer concrete, the more significant the CO_2_ reduction effect. In general, the production of GPC has a lower CO_2_ emission than that of OPC concrete. The results show that for the same compressive strength, GPC can reduce emissions by up to 166.36 kg CO_2_/m^3^ compared to OPC, which is approximately 62.73% of its carbon emissions.

Compared with the results shown in Figure 2, it can be seen that the CO_2_ emissions increased continuously with increasing the compressive strength for OPC. However, for GPC, the CO_2_ emission showed a slight correlation with compressive strength [17], which might be attributed to the fact that the compressive strength and CO_2_ emission of OPC concrete is greatly affected by the amount of cement. Therefore, it can be concluded that GPC can effectively improve the environmental impact without compromising its strength. For geopolymer, the maximum impact on the environment is the alkali activator [18]. Salas [19] showed that the higher the compressive strength, the higher the environmental impacts of higher activator quantities. According to the distributions of CO_2_ emissions in GPC and OPC shown in Figure 2, it can be indicated that the relationship between the different phases is Csc >> Cc > Cys for GPC, while for OPC concrete, it is Csc >> Cys >> Cc. The CO_2_ emission of the production of raw materials is the largest for both.

In Figure 3a, the CO_2_ emission in the raw materials production process of GPC reaches 74–80% of the total emissions, in which sodium silicate takes the most significant proportion, accounting for about 70–84%, which is followed by sodium hydroxide, making up around 10–19%. The commonality in mixes with the lowest CO_2_ emission is in using less alkali activator solution. Therefore, it is suggested to employ sustainable production technology of alkali activator solution and control its usage. Bajpai [20] proposed that silica fume, instead of sodium silicate, can further reduce the environmental impact of geopolymer concrete. On the other hand, considering that the economic value allocation procedure is used to distribute the environmental impact of fly ash, CO_2_ emission of fly ash is generated, accounting for 6–10%. The environmental impact of different aluminosilicate materials varies. Using silica fume instead of fly ash in the preparation of geopolymer concrete can further reduce the environmental impact [20]. The distribution of the CO_2_ emission concerning the aggregates and water shows no striking fluctuation for all mixes. It is clear that the CO_2_ emission of GPC mainly depends on transportation, heat curing, and the alkali activator solution. However, almost all emissions come from cement for OPC concrete, taking up 92–93%. It means that the CO_2_ emission of OPC concrete is affected dramatically by the amount of cement.

Compared with the CO_2_ emissions in concrete production shown in Figure 3, the production of GPC resulted in high CO_2_ emissions due to the high-temperature curing for 24 h, accounting for around 11–15%. It is much higher than OPC concrete, which accounts for 0.2–0.3%. It was observed that when GPC was cured under room temperature similar to OPC concrete, it will further lower the disadvantageous effect on the environment. The CO_2_ emission in the transportation of GPC accounts for about 8–11%, which is not much different from OPC concrete at 7%. It mainly depends on the distance and mode of fly ash and cement transportation. The transport distance has a high impact on the global warming potential of geopolymer mixes [20]. Therefore, it is better to purchase materials close to the plant and choose a transportation mode with low energy consumption. The influence of the materials’ source location and transport mode significantly affect both environmental impacts and production cost and thus should be a significant consideration [9].

### 3.2. Interpretation of MANOVA on GPC

The above results show that the CO_2_ emission of GPC does not increase proportionately and even decreases with the increase in compressive strength. For example, 1 m^3^ C40 GPC emits 336.54 kg CO_2_/m^3^, while C60 and C70 GPC can emit less CO_2_ than C40. Therefore, the growth of the environmental impact should not only be judged by the increase in compressive strength. It can be seen from some studies that various factors are affecting the strength [21,22,23] and CO_2_ emissions of GPC [24]. So, it was necessary to investigate further the correlation of the various impact factors contributing to the compressive strength and CO_2_ emissions.

The orthogonal design method was used in the experimental design stage because it is necessary to simultaneously study the effect of multiple impact factors and save costs/time in testing. Taking the sodium hydroxide concentration (C_NaOH_), sodium silicate to sodium hydroxide ratio (SS/SH), and alkali activator solution to fly ash ratio (S/F) as three variable factors, orthogonal experiments for the three factors and four levels were designed. The test results are shown in Table 4.

The idea of MANOVA is to examine the contribution of different sources of variation to the overall variation based on experimental data to assess each parameter’s importance. Therefore, the F test and significant value were performed. Generally, the parameter change has an essential effect on the experiments when the F value is large or the sig value is close to zero.

Table 5 showed that:(1)Sig < 0.05, R^2^(a) was 0.999 and R^2^(b) was 0.962, which show strong positive correlation. The experimental error was deficient (0.1% and 3.8%), confirming that this model has an excellent fitting effect.(2)According to the sig values, it is concluded that the S/F ratio has a significant influence on CO_2_ emission (Sig = 0.000), which is followed by SS/SH (2 × 10^−6^) and C_NaOH_ (1.6 × 10^−4^). As for compressive strength, C_NaOH_ (Sig = 0.000) has a remarkable impact on it, which is followed by the SS/SH (Sig = 0.008) and S/F ratio (Sig= 0.050).(3)The percentage contribution in Figure 4 confirms the same conclusion.

To further investigate the differences and significance of different levels, Bonferroni and Tukey–Kramer analyses were performed using SPSS.

According to Table 6 and Table 7, increasing C_NaOH_ from 8 to 14 mol/L showed a gradual increase in compressive strength and CO_2_ emission. However, the increase in C_NaOH_ within the range 10–12 mol/L showed no evident difference in the CO_2_ emission. When other C_NaOH_ changed, it showed a stronger and obvious influence. As for compressive strength, when C_NaOH_ varied from 8 mol/L or 10 mol/L to other concentrations, it changed significantly. Meanwhile, the effect of 12 mol/L and 14 mol/L showed a slight difference, which means that the increase in compressive strength slows down after C_NaOH_ reaches 12 mol/L. The compressive strength increases with the increase in concentration of sodium hydroxide. However, beyond a specific range, too much OH^-^ can affect the dissolution of fly ash and thus negatively affect the mechanical properties [25,26].

From Table 6 and Table 7, the effect of SS/SH on CO_2_ emissions showed a big difference with the change of varied ratios within 2 to 4. Meanwhile, CO_2_ emissions increased gradually by increasing the ratio from 2 to 4. On the contrary, as the ratio of SS/SH declined from 4 to 2, the compressive strength gradually increased. Overall, different ratios exerted different influences on the compressive strength. They are mainly embodied in the ratios going from 4 to 2.5 and 4 to 2. However, the effects of the adjoining ratios on compressive strength are not different. Obviously, the smaller the SS/SH ratio or the higher the C_NaOH_ is, the higher the PH and alkalinity of the solutions. Previous studies showed that relatively high alkalinity and pH could provide an alkaline environment conducive to the polymeric reaction and improve the activation effect of fly ash to obtain a compact structure and promote strength growth [27,28,29].

As shown in Table 6, the sig value of CO_2_ emission is exceptionally close to zero, which means that the CO_2_ emission changes significantly as the S/F changes within the range from 0.4 to 0.52. Meanwhile, CO_2_ emission increases as it increases from 0.4 to 0.52. However, the change of S/F showed an insignificant effect on the compressive strength. In addition, its increase will prompt the compressive strength, while when exceeding a certain value (0.48), it will decrease. It might be attributed to the increase in Si/Al ratio due to alkali activator solution usage, leading to a higher compressive strength with a more compact structure [30]. Meanwhile, many studies proved that a too high S/F ratio significantly reduces the compressive strength and the excessive amount of the alkaline activator that leads to inhibition of the geopolymerization process [31,32].

According to MANOVA analysis, the optimum mix based on CO_2_ emission and compressive strength is shown in Table 8.

### 3.3. Interpretation of GRA on GPC

According to Equations (3)–(8), the gray relation coefficient and gray relational degrees of the three parameters to CO_2_ emission and compressive strength are calculated and shown in Table 9. The gray relational degree of compressive strength showed r_1_ (1) > r_1_ (2) > r_1_ (3) > 0.5, indicating that sodium hydroxide concentration exerts a significant impact on the compressive strength, which is followed by the SS/SH ratio and S/F ratio. Concerning CO_2_ emission results, it showed 0.5 < r_2_ (1) < r_2_ (2) < r_2_ (3), confirming that the S/F ratio is the most notable impact factor among the three studied parameters. Thus, the sequences of the gray relational degree of compressive strength and CO_2_ emission results are just the opposite. Therefore, it can be suggested that CO_2_ emissions will not increase continuously by increasing the compressive strength as they do for OPC concrete.

## 4. Conclusions

The present study quantified the CO_2_ emissions of fly ash GPC under different strength grades compared with OPC concrete. It investigated the correlation and influence of different impact factors contributing to CO_2_ emissions and compressive strength based on MANOVA and GRA methods. The following results can be concluded:The CO_2_ emissions from the production of GPC are lower than those of OPC concrete at the same compressive strength. At 70 MPa, the CO_2_ emissions of GPC are reduced by 166.36 kg CO_2_/m^3^, which is approximately 62.73% of its carbon emissions.The CO_2_ emissions of OPC concrete were continuously increasing with the increase in compressive strength. However, there is no significant increase in CO_2_ emissions from high-strength grade GPC. GPC can effectively improve environmental impact without compromising strength.The CO_2_ emission of GPC mainly depends on the alkali activator solution, transportation, and heat curing. However, for OPC concrete, the CO_2_ emissions depend mainly on the amount of cement used.The three studied parameters showed different characteristics and degrees of influence on the compressive strength and CO_2_ emission. For compressive strength, the influence sequence was C_NaOH_ > SS/SH > S/F. However, the effect of CO_2_ emissions was the opposite. Therefore, the mix proportion can be optimized to meet the strength requirements without ignoring the environmental issues.At 12 mol/L C_NaOH_, both the maximum improvement in compressive strength and environmental benefits were guaranteed. This is because the increase in compressive strength slows down with the increase in C_NaOH_ after it reaches 12 mol/L, where its influence on CO_2_ emission was not significant.The effect of SS/SH ratio on CO_2_ emissions and compressive strength was the opposite. Therefore, a lower ratio can obtain higher compressive strength and further reduce CO_2_ emissions.When the workability of GPC is satisfied, the S/F ratio should be reduced as far as possible to meet the environmental benefits.

Overall, conducting comprehensive and in-depth research on the various impact factors prior to determining the mix proportion of GPC could ensure meeting the bilateral consideration of mechanical property and environmental benefits.

## Figures and Tables

**Figure 1 materials-14-07375-f001:**
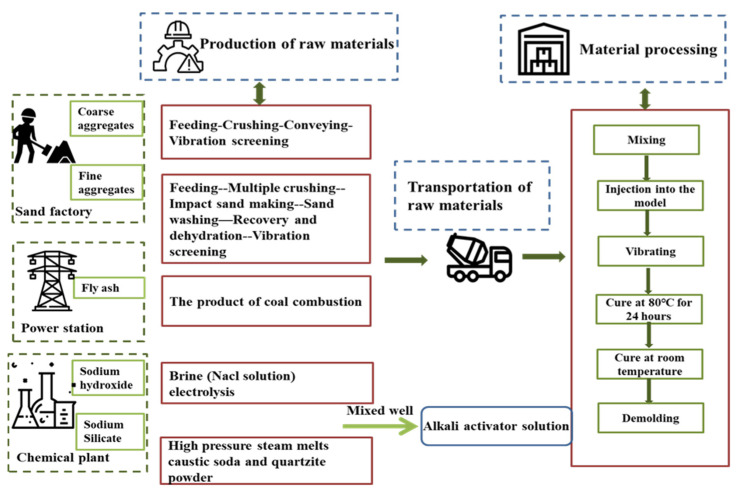
System scope of life cycle assessment.

**Figure 2 materials-14-07375-f002:**
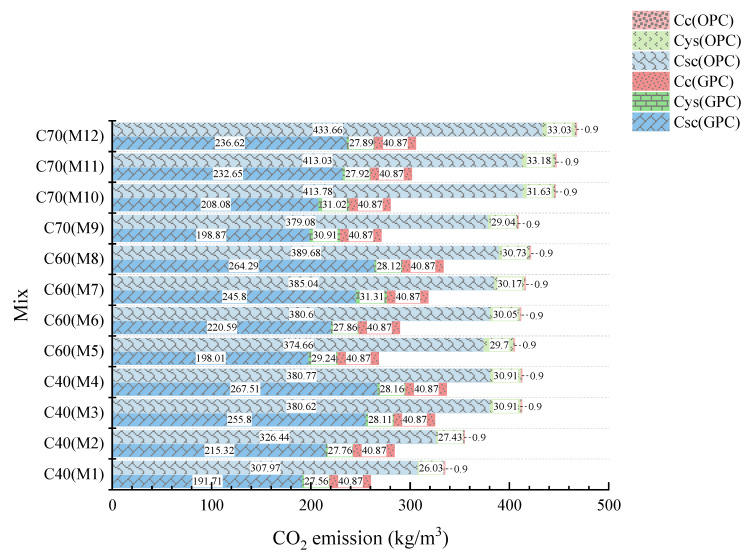
Distribution of CO_2_ emission of different phases. Notes: Csc: Carbon emission of raw materials production. Cys: Carbon emission of transportation. Cc: Carbon emission of concrete production.

**Figure 3 materials-14-07375-f003:**
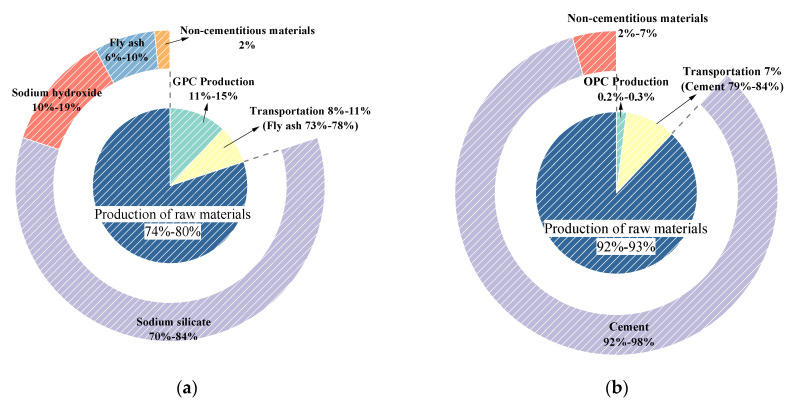
Distributions of CO_2_ emission of ingredients and phases of concrete:(**a**) GPC; (**b**) OPC.

**Figure 4 materials-14-07375-f004:**
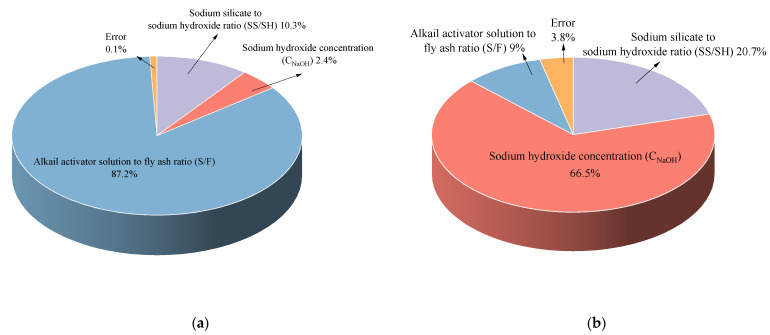
Percentage contribution of the impact factors of results: (**a**) CO_2_ emission; (**b**) CompresScheme

**Table 1 materials-14-07375-t001:** Emission factors of raw materials from the Chinese Life Cycle Database (CLCD).

Materials	Carbon Emission Factor	Units
Fine aggregates	2.820 × 10^−3^	Kg CO_2_/kg
Coarse aggregates	2.440 × 10^−3^	Kg CO_2_/kg
Water	1.891 × 10^−4^	Kg CO_2_/kg
Sodium silicate	1.247	Kg CO_2_/kg
Sodium hydroxide	1.448	Kg CO_2_/kg
OPC (average markets of China)	7.310 × 10^−1^	Kg CO_2_/kg
Concrete reducing water agent	3.000	Kg CO_2_/kg

**Table 2 materials-14-07375-t002:** Emission factors of transportation.

Materials	Means of Transport	Distance	Carbon Emission Factor	Units
Fine aggregates	Medium diesel truck (8 t)	20 km	0.149	Kg CO_2_/tkm
Coarse aggregates	Medium diesel truck (8 t)	20 km	0.149	Kg CO_2_/tkm
Fly ash	Medium diesel truck (8 t)	300 km	0.149	Kg CO_2_/tkm
Sodium hydroxide	Light diesel truck (2 t)	60 km	0.212	Kg CO_2_/tkm
Sodium silicate	Light diesel truck (2 t)	60 km	0.212	Kg CO_2_/tkm
OPC	Medium diesel truck (8 t)	300 km	0.149	Kg CO_2_/tkm

**Table 3 materials-14-07375-t003:** Allocation procedure of fly ash.

Products	Mass	Economic Value Allocation Procedure (Allocation Coefficients)
Electricity (major product)	1 kWh	97%
Fly ash (by-product)	0.109 kg	3%

**Table 4 materials-14-07375-t004:** Mix proportion of fly ash GPC.

Run	C_NaOH_ (mol/L)	SS/SH	S/F	CO_2_ Emission (kg/m^3^)	Compressive Strength (MPa)
1	8	2	0.4	260.05	39.2
2	10	2.5	0.4	271.75	52.6
3	12	3	0.4	280.2	58.0
4	14	4	0.4	289.27	59.1
5	8	2.5	0.44	283.95	43.6
6	10	2	0.44	280.65	64.2
7	12	4	0.44	304.83	48.6
8	14	3	0.44	300.63	71.0
9	8	3	0.48	311.71	37.2
10	10	4	0.48	324.78	40.8
11	12	2	0.48	305.19	73.5
12	14	2.5	0.48	317.88	76.2
13	8	4	0.52	341.31	24.2
14	10	3	0.52	335.34	38.7
15	12	2.5	0.52	332.18	59.2
16	14	2	0.52	329.83	63.2

**Table 5 materials-14-07375-t005:** Test of the inter-subjectivity effect.

	Dependent Variable	Type III Square Sum	Df	Percentage Contribution (%)	F *	Sig
Calibration model	CO_2_ emission	9223.708	9	—	642.460	0.000
	Compressive strength	3217.106	9	—	17.003	0.001
Intercept	CO_2_ emission	1,482,032.325	1	—	929,053.735	0.000
	Compressive strength	45,081.906	1	—	2144.396	0.000
SS/SH	CO_2_ emission	953.381	3	10.33	199.218	2 × 10^−6^
	Compressive strength	692.062	3	20.70	10.973	0.008
C_NaOH_	CO_2_ emission	218.151	3	2.36	45.585	1.6 × 10^−4^
	Compressive strength	2223.612	3	66.51	35.257	0.000
S/F	CO_2_ emission	8052.176	3	87.21	1682.578	0.000
	Compressive strength	301.432	3	9.02	4.779	0.050
Error	CO_2_ emission	9.571	6	0.10	—	—
	Compressive strength	126.139	6	3.77	—	—

Df means degree of freedom. F *: statistical magnitude. Sig represents significant value. Where the Sig is 0.000 means it is close to 0.

**Table 6 materials-14-07375-t006:** Bonferroni analysis simplified results on SPSS

Dependent Variable	C_NaOH_	C_NaOH_	Sig1	SS/SH	SS/SH	Sig2	S/F	S/F	Sig3
CO_2_ emission	8.00	10.00	0.029	2.00	2.50	0.001	0.40	0.44	0.000
		12.00	0.002		3.00	0.000		0.48	
		14.00	0.000		4.00	0.000		0.52	
	10.00	8.00	0.029	2.50	2.00	0.001	0.44	0.40	0.000
		12.00	0.196		3.00	0.005		0.48	
		14.00	0.002		4.00	0.000		0.52	
	12.00	8.00	0.002	3.00	2.00	0.000	0.48	0.40	0.000
		10.00	0.196		2.50	0.005		0.44	
		14.00	0.032		4.00	0.001		0.52	
	14.00	8.00	0.000	4.00	2.00	0.000	0.52	0.40	0.000
		10.00	0.002		2.50	0.000		0.44	
		12.00	0.032		3.00	0.001		0.48	
Compressive strength	8.00	10.00	0.042	2.00	2.50	1.000	0.40	0.44	1.000
		12.00	0.002		3.00	0.209		0.48	1.000
		14.00	0.000		4.00	0.012		0.52	0.712
	10.00	8.00	0.042	2.50	2.00	1.000	0.44	0.40	1.000
		12.00	0.049		3.00	0.511		0.48	1.000
		14.00	0.008		4.00	0.024		0.52	0.105
	12.00	8.00	0.002	3.00	2.00	0.209	0.48	0.40	1.000
		10.00	0.049		2.50	0.511		0.44	1.000
		14.00	0.352		4.00	0.286		0.52	0.102
	14.00	8.00	0.000	4.00	2.00	0.012	0.52	0.40	0.712
		10.00	0.008		2.50	0.024		0.44	0.105
		12.00	0.352		3.00	0.286		0.48	0.102

**Table 7 materials-14-07375-t007:** Tukey–Kramer analysis simplified results on SPSS.

		N	Subset of CO_2_ Emission	Subset of Compressive Strength
C_NaOH_ (mol/L)	8.00	4	299.255	36.050
	10.00	4	303.130	49.075
	12.00	4	305.600	59.825
	14.00	4	309.403	67.375
SS/SH	2.00	4	293.930	60.025
	2.50	4	301.440	57.900
	3.00	4	306.970	51.225
	4.00	4	315.048	43.175
S/F	0.40	4	275.318	52.225
	0.44	4	292.515	56.850
	0.48	4	314.890	56.925
	0.52	4	334.665	46.325

**Table 8 materials-14-07375-t008:** Optimum mix based on CO_2_ emission and compressive strength.

Optimum Mix	C_NaOH_	SS/SH	S/F
CO_2_ emission	8	2	0.40
Compressive strength	14	2	0.48
CO_2_ emission + Compressive strength	12	2	0.40

**Table 9 materials-14-07375-t009:** Gray relational degrees.

Impact Factors	Gray Relational Degree of Compressive Strength (r_1_ (i))	Gray Relational Degree of CO_2_ Emission (r_2_ (i))
C_NaOH_	0.632	0.559
SS/SH	0.622	0.662
S/F	0.616	0.679

## Data Availability

Data are contained within the article or supplementary material or are available on request from the corresponding author.

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
