# Peer review of "Life Cycle Assessment and Impact Correlation Analysis of Fly Ash Geopolymer Concrete"

_materials, 2021, doi:10.3390/ma14237375_

Round 1

Reviewer 1 Report

In the Reviewer opinion the research paper entitled “Life cycle assessment and impact correlation analysis of fly ash geopolymer concrete” is good.

The present study aimed to quantify and compare the environmental impact of fly ash GPC and ordinary Portland cement (OPC) concrete under different strength grades systematically by conducting life cycle assessment (LCA) with examined compressive strength. The CO2 emission of GPC can be reduced by 62.73%, and the correlation between CO2 emission and compressive strength is not significant for GPC. The results indicated that fly ash GPC effectively controls the environmental deterioration without compromising its compressive strength, even in favor of it.

Some comments which greatly enhance the understanding of the paper and its value are presented below. Specific issues that require further consideration are:

  1. The title of the manuscript is matched to its content.
  2. The Introduction generally covers the cases.
  3. The methodology was clearly presented.
  4. In the Reviewer’s opinion, the current state of knowledge relating to the manuscript topic has been presented, but the author's contribution and novelty are not enough emphasized.
  5. Experimental program and results looks interesting and was clearly presented.
  6. In the Reviewer’s opinion, the bibliography, comprising 34 references, is rather representative.
  7. An analysis of the manuscript content and the References shows that the manuscript under review constitutes a summary of the Author(s) achievements in the field.
  8. In the Reviewer’s opinion the manuscript should be published in the journal after major revision.

Author Response

Dear Reviewer:

Thanks for your important comments on our manuscript entitled ‘Life cycle assessment and impact correlation analysis of fly ash geopolymer concrete’ (materials-1449583). The authors appreciate the constructive comments provided by the reviewers. Those comments have been constructive for the authors to improve this manuscript.

Response to Reviewer 1

Comment 1: The title of the manuscript is matched to its content.

Response: Thanks to the reviewers for this comments. The manuscript's title concisely summarizes the content.

Comment 2: The Introduction generally covers the cases.

Response: In the introduction part, revise the review of existing literature to highlight the gaps in the literature and the contribution of this study.

Comment 3: The methodology was clearly presented.

Response: Add the equation for calculating CO2 emissions in ‘2.2.1. Functional unit and system scope’ to improve the reader’s understanding.

Comment 4: In the Reviewer’s opinion, the current state of knowledge relating to the manuscript topic has been presented, but the author's contribution and novelty are not enough emphasized.

Response: The available literature only compares the environmental impact of GPC and OPC for a given compressive strength and does not investigate whether the different parameters in the mix have the same effect on strength and CO2 emissions. However, the combined effect of parameters on mechanical strength and environment should be considered in green concrete design. Accordingly, this study has quantified and compared the environmental impact of fly ash GPC and OPC concrete under different strength grades from manufacturing to production. Using Multivariate Analysis of Variance (MANOVA) and Grey Relational Analysis (GRA) to determine the main impact factors and obtain the correlation of different impact factors contributing to CO2 emissions and compressive strength. Part of the introduction has been revised to highlight the contribution and innovation of this study.

Comment 5: Experimental program and results looks interesting and was clearly presented.

Response: Thanks for the reviewer’s positive comments. The results of the trial have been discussed in more detail.

Comment 6: In the Reviewer’s opinion, the bibliography, comprising 34 references, is rather representative.

Response: Some of the most recent literature in the field has been added, confirming the conclusions drawn in this study.

Comment 7: An analysis of the manuscript content and the References shows that the manuscript under review constitutes a summary of the Author(s) achievements in the field.

Response: Thank you for your comments.

Comment 8: In the Reviewer’s opinion the manuscript should be published in the journal after major revision.

Response: The manuscript has been revised as requested by the reviewers, and editorial errors in the manuscript have been corrected to improve readability and editorial quality.

Reviewer 2 Report

Thank you for the invitation to review the paper entitled “Life cycle assessment and impact correlation analysis of fly ash geopolymer concrete”. The manuscript is interesting, but there are several issues required to be addressed: 

  1. In abstract: “..The significance of the effect of the three factors on the com-pressive strength is CNaOH>SS/SH>S/F, the effect of CO2 emission was completely opposite. In this way, in addition to the cost-effective ratio of one of the products, another environmental burden index was added. The results indicated that fly ash GPC effectively controls the environmental deterioration without compromising its compressive strength, even in favor of it.” These are like general known results. They should be more specific by adding some percentage
  2. In the Introduction part, the authors need to narrow down the review towards the problem, highlighting the gaps in literature and ending with defining the problem statements and the objective of this research.
  3. The authors mentioned “All these studies came to a similar conclusion that GPC has lower CO2 emissions compared to OPC concrete under a certain compressive strength. Almost all the quantitative studies of GPC were based on a single compressive strength, without a comparative analysis of GPC and OPC concrete under different compressive strengths. Thus, compressive strength was used as the only or primary standard for mix proportion design, ignoring the importance of the environmental impact.” However, they only used three different compressive strengths and not a wide range of compressive strengths. For a better comparison it would be better to have a wider range of different strengths. Also, please explain what was the reason for the strength selection as C40, 60 and 70.
  4. In Materials and Methods, please provide a better explanation of how the mixes are prepared and casted.
  5. The discussion part needs to be improved. The authors should compare and present more comments about the findings of these experiments. It is suggested to evaluate other aspects of work as the different effects between your materials and other researches. The results of the current study are better to be elaborated and confirmed by the findings of studies available in literature.
  6. Please improve the write up in line 289.
  7. In Table S3, it’s not obvious which mixes are related to each strength. Please revise it.

Author Response

Response to reviewers’ comments

Manuscript Title: Life cycle assessment and impact correlation analysis of fly ash geopolymer concrete

Authors: Xiaoshuang Shi, Cong Zhang, Yongchen Liang, Jinqian Luo, Xiaoqi Wang, Ying Feng, Yanlin Li, Qing-yuan Wang, and Abd El-Fatah Abomohra

Manuscript Number: materials-1449583

Article Type: Article

Dear Reviewer:

Thanks for your important comments on our manuscript entitled ‘Life cycle assessment and impact correlation analysis of fly ash geopolymer concrete’ (materials-1449583). The authors appreciate the constructive comments provided by the reviewers. Those comments have been constructive for the authors to improve this manuscript.

Response to Reviewer 2

Comment 1: In abstract: “..The significance of the effect of the three factors on the com-pressive strength is CNaOH>SS/SH>S/F, the effect of CO2 emission was completely opposite. In this way, in addition to the cost-effective ratio of one of the products, another environmental burden index was added. The results indicated that fly ash GPC effectively controls the environmental deterioration without compromising its compressive strength, even in favor of it.” These are like general known results. They should be more specific by adding some percentage

Response: The corresponding narrative in the abstract has been amended to read ‘The results indicated that the CO2 emission of GPC can be reduced by 62.73%, and the correlation between CO2 emission and compressive strength is not significant for GPC. The degree of influence of the three factors on the compressive strength is CNaOH (66.5%)>SS/SH (20.7%)>S/F (9%) and on CO2 emissions is S/F (87.2%)> SS/SH (10.3%)> CNaOH (2.4%).’ Added percentage contribution after each factor.

Comment 2: In the Introduction part, the authors need to narrow down the review towards the problem, highlighting the gaps in literature and ending with defining the problem statements and the objective of this research.

Response: The review in the introduction has been appropriately added or subtracted to highlight gaps in the existing literature, and revise the statement in the final part of the introduction.

Comment 3: The authors mentioned “All these studies came to a similar conclusion that GPC has lower CO2 emissions compared to OPC concrete under a certain compressive strength. Almost all the quantitative studies of GPC were based on a single compressive strength, without a comparative analysis of GPC and OPC concrete under different compressive strengths. Thus, compressive strength was used as the only or primary standard for mix proportion design, ignoring the importance of the environmental impact.” However, they only used three different compressive strengths and not a wide range of compressive strengths. For a better comparison it would be better to have a wider range of different strengths. Also, please explain what was the reason for the strength selection as C40, 60 and 70.

Response: The strength of fly ash geopolymer concrete in the literature is usually in the range of 40 to 70 MPa. Combined with our group's previous research, a representative mix ratio was selected for concrete compressive strength testing. From the test results, the strengths were determined to be C40, C60, and C70. The mix proportions of OPC of different strengths are determined according to the specifications, and specimens are cast and tested for compressive strength.

Comment 4: In Materials and Methods, please provide a better explanation of how the mixes are prepared and casted.

Response: In Materials and Methods, a detailed description of the preparation and pouring method of the mix is given.

‘After all the materials were prepared, GPC and OPC of different strengths were designed based on our previous studies, and 100 x 100 x 100mm specimens were pre-pared for compressive strength testing ( Table S3 and S4). Firstly, dry mix the fine and coarse aggregate in the mixer for 1min, then pour into the fly ash and mix for another 1min. Secondly, add the alkali activator solution and continue mixing for 2min. Finally, the fresh concrete is placed in steel molds and compacted on a vibrator. In order to promote the development of compressive strength, GPC was covered with plastic film to prevent moisture loss and cured at 80 ℃ for 24 hours and then cured at room temperature for 27 days. OPC was cured for 28 days under standard curing conditions.’

Comment 5: The discussion part needs to be improved. The authors should compare and present more comments about the findings of these experiments. It is suggested to evaluate other aspects of work as the different effects between your materials and other researches. The results of the current study are better to be elaborated and confirmed by the findings of studies available in literature.

Response: The discussion part has been improved and the results obtained from the experiment have been confirmed using results from the existing literature.

For Example, The findings of Bajpai's study have been added to 3.1 Interpretation and comparison of GPC and OPC concrete.

‘Bajpai proposed that silica fume, instead of sodium silicate, can further reduce the environmental impact of geopolymer concrete.’(Line2 263-265)

‘The environmental impact of different aluminosilicate materials varies. Using silica fume instead of fly ash in the preparation of geopolymer concrete can further reduce the environmental impact.’(Lines 267-269)

‘The transport distance has a high impact on the global warming potential of geopolymer mixes.’ (Lines 282-283)

Comment 6: Please improve the write up in line 289.

Response: Amend the statement in line 289 to read ‘However, beyond a specific range, too much OH- can affect the dissolution of fly ash and thus negatively affect the mechanical properties.’

Comment 7: In Table S3, it’s not obvious which mixes are related to each strength. Please revise it.

Response: We have modifed tables S3 and S4 so that the mix and strength correspond significantly. The modified Table S3 and Table S4 are shown below:

Table S3 Mix proportion of fly ash GPC under different strength grades (kg/m3)

Strength grades

Mix

CNaOH

(mol/L)

SS/SH

S/F

CA

S

FA

Water

NaOH

Na2SiO3

C40

G4M1

8.00

2.00

0.40

1212.00

544.00

460.00

45.30

16.10

122.70

G4M2

8.00

2.50

0.44

1201.00

539.00

460.00

42.20

15.00

142.90

G4M3

10.00

4.00

0.48

1186.00

533.00

460.00

30.30

13.90

176.60

G4M4

10.00

3.00

0.52

1174.00

527.00

460.00

40.00

18.30

180.90

C60

G6M5

12.00

1.50

0.38

1180.00

530.00

500.00

48.60

27.40

114.00

G6M6

14.00

4.00

0.40

1212.00

544.00

460.00

21.90

14.90

147.20

G6M7

12.00

2.50

0.41

1132.00

508.00

540.00

40.20

22.70

157.10

G6M8

12.00

2.50

0.52

1174.00

527.00

460.00

43.70

24.70

170.90

C70

G7M9

12.00

1.50

0.35

1152.00

518.00

540.00

48.56

27.44

114.00

G7M10

12.00

2.00

0.35

1152.00

518.00

540.00

40.47

22.86

126.67

G7M11

14.00

3.00

0.44

1201.00

539.00

460.00

29.09

19.72

151.30

G7M12

12.00

2.00

0.48

1187.00

533.00

460.00

46.86

26.47

146.67

Table S4 Mix proportion of OPC  concrete under different strength grades (kg/m3)

Strength grades

Mix

CA

S

Cement

Water

Concrete reducing 

water agent

C40

O4M1

1215.00

572.00

415.00

166.00

0.00

O4M2

1104.00

736.00

440.00

170.00

0.00

O4M3

1301.00

481.00

513.00

205.00

0.35

O4M4

1301.00

481.00

513.00

205.00

0.40

C60

O6M5

1198.00

617.00

486.00

170.00

4.90

O6M6

1151.00

647.00

494.00

158.00

4.94

O6M7

1044.00

696.00

500.00

160.00

5.00

O6M8

1145.00

618.00

510.00

163.00

4.10

C70

O7M9

1125.00

633.00

475.00

166.00

9.00

O7M10

1104.00

595.00

531.00

170.00

7.00

O7M11

1250.00

670.00

550.00

210.00

2.00

O7M12

1098.00

652.00

556.00

156.00

7.51

Reviewer 3 Report

Dear Author,

The paper "Life cycle assessment and impact correlation analysis of fly ash 2
geopolymer concrete" is  good in terms of its relevance. Many places spelling mistakes (eg line 94) are there. It needs a through revising in terms of data presentation as well as description.

How is the composition of the fly ash (presented in Table S2) analysed. it would be good to include the errors in measurements if it has been done more than once. If not done, please analyse it again.

A more clear description of the determination of CO2 emission will be good for the readers to understand.

Overall the manuscript needs a thorough editing.

Author Response

Response to reviewers’ comments

Manuscript Title: Life cycle assessment and impact correlation analysis of fly ash geopolymer concrete

Authors: Xiaoshuang Shi, Cong Zhang, Yongchen Liang, Jinqian Luo, Xiaoqi Wang, Ying Feng, Yanlin Li, Qing-yuan Wang, and Abd El-Fatah Abomohra

Manuscript Number: materials-1449583

Article Type: Article

Dear Reviewer:

Thanks for your important comments on our manuscript entitled ‘Life cycle assessment and impact correlation analysis of fly ash geopolymer concrete’ (materials-1449583). The authors appreciate the constructive comments provided by the reviewers. Those comments have been constructive for the authors to improve this manuscript.

Response to Reviewer 3

Comment 1: Many places spelling mistakes (eg line 94) are there. It needs a through revising in terms of data presentation as well as description.

Response: Amend line 94 to read ‘The fly ash was class F low calcium fly ash with an average diameter of 1.586 µm. The characteristics of aggregates (Table S1) and Fly ash (Table S2) are provided in the supplementary data.’ We have comprehensively revisied the representation and description of manuscript data.

Comment 2: How is the composition of the fly ash (presented in Table S2) analysed. it would be good to include the errors in measurements if it has been done more than once. If not done, please analyse it again.

Response: The oxide composition of fly ash, as determined by X-ray fluorescence (XRF), is shown in Table S2. The difference between multiple measurements of the same fly ash is slight, and the error is not expressed in the results in the literature.

Comment 3:A more clear description of the determination of CO2 emission will be good for the readers to understand.

Response: Add the equation for calculating CO2 emissions in ‘2.2.1. Functional unit and system scope’ to improve the reader’s understanding.

where the left-hand side of the equation indicates the amount of CO2 emission (kg CO2/m3) for every cubic meter of concrete, indicates the fraction of component i, and specifies the CO2 emissions (kg).

Reviewer 4 Report

  1. This paper evaluates the LCA and environmental impacts of fly ash-based geopolymer concrete. Authors have evaluated important aspects of GPC and concluded interesting results. However, the writing of the manuscript failed to meet the publication standards and thus, major editorial revisions are needed before considering for publication as per comments below.
  2. The English language and editorial quality of the manuscript are good overall. However, some minor typo errors cab be found and need to be corrected. Below are some examples:
  • Lines 42 and 43: The reference given at the end of a sentence or phrase should be spaced from the last word. For example, consumption[1] should be changed to consumption [1]. The same error is repeated in the entire manuscript.
  • In some parts of the manuscript, the sentences are very long. The reviewer kindly asks the authors to revise this problem. For instance, in line 62, the sentence should be finished after “OPC concrete” with a period as below:

….of OPC concrete. The reason for…

  • Some typo mistakes need to be revised. In line 95, “suppletory” should be changed to “supplementary”.

These are only a few examples of the editorial mistakes that can be found in the manuscript. Therefore, authors, with the help of a native English writer or expert, should carefully revise this manuscript to improve the readability and editorial quality.

  1. The font size of the texts on Figures 2 and 3 is very small so that is hard to read the numbers. The figures should be revised for this problem.
  2. Lines 196 195: It is concluded that the reduction in CO2 emissions is more significant in higher compressive strength. How authors can justify this conclusion? Is there any similar finding in the literature?
  3. As the reviewer proceed with reading the manuscript, the problem of long sentences without proper conjunction became more evident. The reviewers must resolve this problem in the manuscript to improve the readability. Lines 200-210 are a poorly written part of the manuscript.
  4. One important missing part in this manuscript is the lack of a comprehensive literature review. There have been multiple studies on the LCA and CO2 emissions of alkali-activated materials and geopolymer concrete. Authors are kindly asked to better analyze and review such studies and compare their founding and conclusions with similar studies. Below are a few of them.
  • Nehdi, M.L. and Yassine, A., 2020. Mitigating Portland Cement CO2 Emissions Using Alkali-Activated Materials: System Dynamics Model. Materials13(20), p.4685.
  • Faridmehr, I., Nehdi, M.L., Nikoo, M., Huseien, G.F. and Ozbakkaloglu, T., 2021. Life-Cycle Assessment of Alkali-Activated Materials Incorporating Industrial Byproducts. Materials14(9), p.2401.
  • Bajpai, R., Choudhary, K., Srivastava, A., Sangwan, K.S. and Singh, M., 2020. Environmental impact assessment of fly ash and silica fume based geopolymer concrete. Journal of Cleaner Production, 254, p.120147.
  1. The manuscript has acceptable technical content but needs major editorial revisions. Therefore, it is recommended for publication after addressing the above comments and careful revision for the English language.

Author Response

Response to reviewers’ comments

Manuscript Title: Life cycle assessment and impact correlation analysis of fly ash geopolymer concrete

Authors: Xiaoshuang Shi, Cong Zhang, Yongchen Liang, Jinqian Luo, Xiaoqi Wang, Ying Feng, Yanlin Li, Qing-yuan Wang, and Abd El-Fatah Abomohra

Manuscript Number: materials-1449583

Article Type: Article

Dear Reviewer:

Thanks for your important comments on our manuscript entitled ‘Life cycle assessment and impact correlation analysis of fly ash geopolymer concrete’ (materials-1449583). The authors appreciate the constructive comments provided by the reviewers. Those comments have been constructive for the authors to improve this manuscript.

Response to Reviewer 4

Comment 1:The English language and editorial quality of the manuscript are good overall. However, some minor typo errors cab be found and need to be corrected. Below are some examples:

Lines 42 and 43: The reference given at the end of a sentence or phrase should be spaced from the last word. For example, consumption[1] should be changed to consumption [1]. The same error is repeated in the entire manuscript.

Response: Thanks for your carefully reviewing. The type of question that appears in the manuscript has been revised.

In some parts of the manuscript, the sentences are very long. The reviewer kindly asks the authors to revise this problem. For instance, in line 62, the sentence should be finished after “OPC concrete” with a period as below:

….of OPC concrete. The reason for…

Response: Thanks to the reviewer's suggestions, the long sentences in the article have been revised.

Some typo mistakes need to be revised. In line 95, “suppletory” should be changed to “supplementary”.

These are only a few examples of the editorial mistakes that can be found in the manuscript. Therefore, authors, with the help of a native English writer or expert, should carefully revise this manuscript to improve the readability and editorial quality.

Response: The manuscript has been read carefully, editorial errors in it have been corrected, and readability and editorial quality have been improved with the help of native English experts.

Comment 2:The font size of the texts on Figures 2 and 3 is very small so that is hard to read the numbers. The figures should be revised for this problem.

Response: We have modified the texts in Figures 2 and 3 to improve the readability of the graphical information.

Figure 2. Distribution of CO2 emission of different phases

(a)

(b)

Figure 3. Distributions of CO2 emission of ingredients and phases of concrete:(a) GPC; (b) OPC

Comment 3:Lines 196 195: It is concluded that the reduction in CO2 emissions is more significant in higher compressive strength. How authors can justify this conclusion? Is there any similar finding in the literature?

Response: The difference in CO2 emissions between GPC and OPC at different strengths can be seen in Figure 1, where the higher the compressive strength of GPC, the more significant the reduction in CO2 emissions compared to OPC. CO2 emissions from GPC at 40MPa, 60MPa, and 70MPa are reduced by 20.48%, 27%, and 34.6% respectively compared to OPC. A comparison of GPC and OPC CO2 emissions at different strengths is not currently available in the literature.

Comment 4:As the reviewer proceed with reading the manuscript, the problem of long sentences without proper conjunction became more evident. The reviewers must resolve this problem in the manuscript to improve the readability. Lines 200-210 are a poorly written part of the manuscript.

Response: The problem of long sentences without proper conjunction has been addressed in the manuscript to improve readability.

Amend lines 200-210 to read as follows:

Compared with the results shown in Figure 2, it can be seen that the CO2 emissions increased continuously with increasing the compressive strength for OPC. However, for GPC, the CO2 emission showed a slight correlation with compressive strength [18], which might be attributed to the fact that the compressive strength and CO2 emission of OPC concrete is greatly affected by the amount of cement. Therefore, it can be concluded that GPC can effectively improve the environmental impact without compromising its strength. For geopolymer, the maximum impact on the environment is the alkali activator. Salas [19] showed that the higher the compressive strength, the higher the environmental impacts of higher activator quantities. According to the distributions of CO2 emissions in GPC and OPC showed in Figure 2, it can be indicated that the relationship between the different phases is Csc >> Cc > Cys for GPC, while for OPC concrete is Csc >> Cys >> Cc. The CO2 emission of the production of raw materials is the largest for both.

Comment 5:One important missing part in this manuscript is the lack of a comprehensive literature review. There have been multiple studies on the LCA and CO2 emissions of alkali-activated materials and geopolymer concrete. Authors are kindly asked to better analyze and review such studies and compare their founding and conclusions with similar studies.

Response: Authors are grateful to the reviewer for recommending the literatures, conducting a more comprehensive literature review by reading the relevant literature, and comparing the findings of this paper with similar studies.

In the Introduction part, adds Faridmehr to the expansion of boundary conditions in the study of LCA in AAM (Lines 84-93).

The findings of Bajpai's study have been added to 3.1 Interpretation and comparison of GPC and OPC concrete, for example, ‘Bajpai proposed that silica fume, instead of sodium silicate, can further reduce the environmental impact of geopolymer concrete.’(Line2 263-265) ‘The environmental impact of different aluminosilicate materials varies. Using silica fume instead of fly ash in the preparation of geopolymer concrete can further reduce the environmental impact.’(Lines 267-269) ‘The transport distance has a high impact on the global warming potential of geopolymer mixes.’ (Lines 282-283)

Comment 6:The manuscript has acceptable technical content but needs major editorial revisions. Therefore, it is recommended for publication after addressing the above comments and careful revision for the English language.

Response: Thanks to the reviewers' comments, significant editorial changes have been made, and the issues raised by the reviewers have been addressed.

Round 2

Reviewer 1 Report

Authors corrected articel folow to my sugestion. In my opinion should be published in the Journal.

Reviewer 2 Report

The revisions are made based on the reviewer's comments. The manuscript can be accepted in its present form.

Reviewer 3 Report

The manuscript in the current format  could be accepted for publication after minor spell and grammar check .